# Constitutive Occurrence of E:N-cadherin Heterodimers in Adherens Junctions of Hepatocytes and Derived Tumors

**DOI:** 10.3390/cells11162507

**Published:** 2022-08-12

**Authors:** Tiemo Sven Gerber, Dirk Andreas Ridder, Mario Schindeldecker, Arndt Weinmann, Diane Duret, Kai Breuhahn, Peter R. Galle, Peter Schirmacher, Wilfried Roth, Hauke Lang, Beate Katharina Straub

**Affiliations:** 1Institute of Pathology, University Medical Center of the Johannes Gutenberg-University Mainz, 55131 Mainz, Germany; 2Tissue Biobank, University Medical Center of the Johannes Gutenberg-University Mainz, 55131 Mainz, Germany; 31st Department of Internal Medicine, Gastroenterology and Hepatology, University Medical Center of the Johannes Gutenberg-University Mainz, 55131 Mainz, Germany; 4Institute of Pathology, University Clinic Heidelberg, 69120 Heidelberg, Germany; 5Department of General, Visceral and Transplant Surgery, University Medical Center of the Johannes Gutenberg-University Mainz, 55131 Mainz, Germany

**Keywords:** cadherin, liver tumors, cell-cell contacts, adherens junctions, epithelial–mesenchymal transition

## Abstract

Cell–cell junctions are pivotal for embryogenesis and tissue homeostasis but also play a major role in tumorigenesis, tumor invasion, and metastasis. E-cadherin (*CDH1*) and N-cadherin (*CDH2*) are two adherens junction’s transmembrane glycoproteins with tissue-specific expression patterns in epithelial and neural/mesenchymal cells. Aberrant expression has been implicated in the process of epithelial–mesenchymal transition (EMT) in malignant tumors. We could hitherto demonstrate cis-E:N-cadherin heterodimer in endoderm-derived cells. Using immunoprecipitation in cultured cells of the line PLC as well as in human hepatocellular carcinoma (HCC)-lysates, we isolated E-N-cadherin heterodimers in a complex with the plaque proteins α- and β-catenin, plakoglobin, and vinculin. In confocal laser scanning microscopy, E-cadherin co-localized with N-cadherin at the basolateral membrane of normal hepatocytes, hepatocellular adenoma (HCA), and in most cases of HCC. In addition, we analyzed E- and N-cadherin expression via immunohistochemistry in a large cohort of 868 HCCs from 570 patients, 25 HCA, and respective non-neoplastic liver tissue, and correlated our results with multiple prognostic markers. While E- or N-cadherin were similarly expressed in tumor sites with vascular invasion or HCC metastases, HCC with vascular encapsulated tumor clusters (VETC) displayed slightly reduced E-cadherin, and slightly increased N-cadherin expression. Analyzing The Cancer Genome Atlas patient cohort, we found that reduced mRNA levels of *CDH1*, but not *CDH2* were significantly associated with unfavorable prognosis; however, in multivariate analysis, *CDH1* did not correlate with prognosis. In summary, E- and N-cadherin are specific markers for hepatocytes and derived HCA and HCC. E:N-cadherin heterodimers are constitutively expressed in the hepatocytic lineage and only slightly altered in malignant progression, thereby not complying with the concept of EMT.

## 1. Introduction

Hepatocellular carcinoma (HCC) constitutes the most common primary liver tumor, the sixth most common cancer, and the fourth most common cause of cancer-related death worldwide [1]. In most cases, HCC develops in the context of liver cirrhosis based on chronic liver injury due to alcohol and/or chronic infection by hepatitis viruses. Due to impaired liver function, patients with HCC thereby exhibit limited therapeutic options. In the case of early-stage HCC and good liver function, locoregional curative therapies provide a 5-year survival of more than 70%. In contrast, patients with advanced HCC in combination with poor liver function usually receive palliative systemic therapy with a median survival of approximately 1–1.5 years [2]. Characteristically, HCCs are highly vascularized with nodular infiltrative growth patterns and frequent hemangioinvasion, mostly in the absence of prominent stroma reaction. While the molecular features of HCC have been widely studied, the mechanisms underlying specific HCC cell biology remain largely unclear.

Cell-cell junctions play a critical role in maintaining cell and tissue polarity and integrity. Functionally, cell–cell junctions may be divided into the large subgroups gap junctions, which are essential for cell communication, tight junctions sealing the apical from the basolateral cell membrane, and adhering junctions acting as mechanical tethers that also organize the cytoskeleton and influence tissue interaction (for an overview, see the seminal publication of Farquhar and Palade [3]). In contrast to desmosomes, which are linked to the intermediate filament cytoskeleton in epithelia, adherens junctions (AJs) are more widely found and associated with the actin cytoskeleton [4,5,6]. AJs contain calcium-dependent transmembrane glycoproteins of the cadherin superfamily with cell- and tissue-specific expression patterns [7]. The extracellular domain of cadherins is believed to interact in a homologous fashion with the identical cadherin from the neighboring cell, thereby mediating cell sorting and tissue organization. E-cadherin (*CDH1*, formerly known as LCAM or uvomorulin [8,9]) is specific to epithelial cells, whereas N-cadherin (*CDH2*, formerly known as ACAM) is expressed in neuroepithelial and mesenchymal cells [10,11]. Both cadherins are well conserved among the species. The intracellular cadherin domain is linked to the actin cytoskeleton via the cytoplasmic plaque proteins α- and β-catenin (*CTNNA1* and *CTNNB1*), plakoglobin (*JUP*), p120 catenin (p120-ctn; *CTNND1*), and vinculin (*VCL*). Through this interaction, also intra- and extracellular signaling, nuclear and transcriptional functions, and cell homeostasis is regulated. α- and β-catenin, as well as p120-ctn, bind to E-cadherin, which protects E-cadherin from degradation [6,12,13]. In addition, via β-catenin, AJs have been linked to Wnt-signaling, which also plays a major role in hepatocarcinogenesis, as up to 30% of HCCs display driver mutations in *CTNNB1* [12,13,14,15,16,17]. At a molecular level, E-cadherin and N-cadherin share several similarities [18]. Both E- and N-cadherin contain five extracellular cadherin domains, which are less conserved than the intracellular domains. While cadherins usually interact in a homodimeric fashion on the lateral or contralateral/adhesive side [19], heterodimeric interactions were also reported [20,21,22]. Interestingly, the loss of a singular junction protein such as β-catenin, E-cadherin, or N-cadherin leads to impaired embryonic development in mice [15,16,17]. Mice with a cardiac knock-in of E-cadherin in the locus of N-cadherin are rescued from cardiac death, yet they develop cardiomyopathy [23,24]. Heterozygous mice with whole-body knock-in of N-cadherin leading to coexpression of E-cadherin and N-cadherin are viable without abnormalities [25]; however, knock-in of N-cadherin in E-cadherin knock-out mice shows impaired embryonic development similar to E-cadherin knock-out mice, although, no effects on tumorigenesis have been observed in these models. 

During the development of multicellular organisms, a change in tissue structure is often associated with changes in cadherin expression regulating cell sorting [26,27]. Importantly, the switch from E-cadherin to N-cadherin is part of the separation process of the neural tube from the embryonic ectoderm layer [28]. During tissue development and embryogenesis, the degree of selective cadherin binding depends on their expression level and affinity partner [29]. While the loss of cell adhesion, e.g., due to mutations of *CDH1*, is a hallmark of certain cancers, tissue-specific cadherin expression is useful to distinguish between different cell types and determine tumor progeny. For instance, lost or aberrant membranous expression of E-cadherin as determined via immunohistochemistry are routinely used as diagnostic markers for lobular invasive carcinoma of the breast [30], and signet cell carcinoma of the stomach [31]. In addition, cadherins are linked to other pathologic processes. For instance, E-cadherin is transcriptionally regulated by the human T-cell leukemia virus type I [32]. In diverse cancers, downregulation of E-cadherin and upregulation of N-cadherin are implicated in malignant progression and unfavorable patient outcomes. The mechanisms are manifold and include enhanced tumor cell invasion, hemangioinvasion, as well as lymph node and foreign metastases facilitated by a process termed epithelial to mesenchymal transition (EMT) [26].

We have previously shown that under physiological conditions, hepatocytes harbor equal amounts of N- and E-cadherin, which are localized in a special type of AJs forming cis-E:N-cadherin heterodimers at the basolateral membrane. Heterodimeric E-N-cadherin complexes may therefore be a characteristic feature of endoderm-derived cells [22]. Thus, hepatocytes may represent an exception to the mutual exclusive E- and N-cadherin pattern observed in embryonic development and tissue formation; however, this also raises the question of whether the increase in N-cadherin over E-cadherin and its assumed prognostic potential of invasive and metastatic biological behavior as implied by the EMT concept, truly also apply to HCC. Until now, a comprehensive analysis of the expression pattern of HCA, HCC, and associated normal liver tissue has not been conducted. 

This study aims to investigate E- and N-cadherin in hepatocarcinogenesis in situ and test their robustness as markers of the hepatocytic lineage. In addition, we determine the prognostic impact of E- and N-cadherin expression in the context of the well-known concept of EMT, although a clear definition of the process of EMT is currently lacking. We systematically and comprehensively characterized E- and N-cadherin tissue expression in a large cohort of patients in normal liver, focal nodular hyperplasia (FNH), dysplastic nodules (DN), HCA, HCC, as well as respective cell cultures using protein biochemistry, immunohistochemistry, and laser scanning immunofluorescence microscopy.

## 2. Materials and Methods

### 2.1. Immunofluorescence Microscopy

Immunofluorescence microscopy was performed as described before [22]. Cryosections of the normal liver as well as of HCC and HCA were cut at a thickness of 5 µm, air-dried for 1 h, and fixed with acetone at −20 °C for 10 min. After a permeabilization step for 4 min in 0.1% triton-X-100, and two washing steps in phosphate-buffered saline (PBS), the primary antibodies were applied for 30 min to 1 h, followed by two washing steps for 5 min in PBS and 30 min incubation with the respective secondary antibodies in a humid chamber (cy 3, rabbit, Dianova; Alexa 488 anti-mouse, MoBiTec GmbH, Göttingen, Germany). After two subsequent washing steps for 5 min in PBS as well as a short washing step in distilled water, slides were dehydrogenated with 100% ethanol for 5 min and mounted with a DAPI embedding medium. A confocal laser scanning immunofluorescence microscope (LSM 510 Meta; Carl Zeiss AG, Oberkochen, Germany) equipped with Plan Apochromat 63×/1.40 NA oil and Plan-Neofluar 40×/1.30 NA oil objectives was used. AxioVision Release 4.6.3.0 and LSM Image browser 3.2.0.115 software (Carl Zeiss AG, Oberkochen, Germany) was used for image processing.

### 2.2. Antibodies and Reagents

For primary antibodies used in this study, see Appendix A. Secondary antibodies used were Alexa 488- and 594-coupled mouse and rabbit antibodies (MoBiTec, Göttingen, Germany) as well as the respective cy3-coupled mouse, rabbit, and guinea pig antibodies (Dianova, Hamburg, Germany) and diluted 1:200 up to 1:500 according to the manufacturer’s instructions.

### 2.3. Immunoblot Analysis

SDS-PAGE was performed as described [33]. Samples were taken up in a sample buffer, with the addition of benzonase to the initial buffer (1:1000; Merck, Darmstadt, Germany). Immunoblotting was then performed using PVDF membranes (Millipore, Bedford, MA, USA). After blocking with 10% non-fat dry milk in Tris-buffered saline containing 0.1% Tween (TBST) for at least 1 h, blots were incubated with the specific primary antibody solution in PBS for 1 h, followed by three washes in TBST for 30 min each. Horseradish peroxidase (HRP)-conjugated secondary antibodies to rabbit, mouse, or guinea pig IgG (diluted 1:10,000 in TBST) were then applied for 30 min, followed by 30 min washes in TBST and a short incubation in enhanced chemiluminescence solution (ECL; Amersham Biosciences, Freiburg, Germany).

### 2.4. Immunoprecipitation

HCC tissue pieces or cultured cells were taken up in Triton-X-100 containing immunoprecipitation buffer (“Triton-IP-buffer”; 20 mM Tris-HCl, pH 7.5, 150 mM NaCl, 5 mM EDTA or 0.5 mM CaCl2, 1% Triton X-100, 1 mM DTT, complete protease inhibitors), in RIPA immunoprecipitation buffer (“RIPA-IP-buffer”; 20 mM HEPES, pH 7.4, 150 mM NaCl, 5 mM EDTA or 0.5 mM CaCl2, 1% Triton, 0.5% sodium deoxycholate, 0.1% SDS, 1 mM DTT, protease inhibitors) and in Empigen-containing immunoprecipitation buffer (“Empigen-IP-buffer”; 20 mM Tris-HCl, pH 7.5, 150 mM NaCl, 5 mM EDTA or 0.5 mM CaCl2, 0.1% or 0.5% Empigen, 1 mM DTT, protease inhibitors) and centrifuged at 15,000 rpm in a laboratory centrifuge 5414 (Eppendorf; Hamburg, Germany) and 4 °C for 15 min. The supernatant obtained was then precleared with protein G- or A-coupled magnetic beads (Dynal Dynabeads, Invitrogen, Karlsruhe, Germany) for several hours, and in parallel, protein A and/or protein G magnetic beads were coated with the specific antibody and with an unrelated antibody (mouse myeloma IgG1, Zymed/Thermo Fisher Scientific, Waltham, MA, USA) in 50 mM Tris-HCl, pH 7.5 at 4 °C for several hours. The precleared supernatant was then incubated with the antibody-coupled beads at 4 °C overnight and washed intensely afterward. The pellets obtained were solubilized in 20–40 μL sample buffer, and the immunoprecipitates were compared with the precleared pellets and the supernatants before and after IP by SDS-PAGE.

### 2.5. Tissues and Cells

Tissue samples from 868 HCCs of 570 patients that underwent HCC resection at the University Medical Center Mainz from 1997 to 2017 were provided by the Tissue Biobank of the University Medical Center Mainz. Clinical data of HCC patients were retrieved from a prospectively populated clinical database [34]. HCC, HCA, and DN were reviewed and staged using the criteria of the WHO classification of tumors of the digestive system (5th edition, 2019) by two experienced hepatopathologists (DAR, BKS, compare [34,35,36]). Tissue microarray (TMA) blocks were generated with HCC, HCA, DN, and respective non-neoplastic liver tissue. TMA cores that could not be evaluated due to loss of tissue during processing were excluded from this study. HCC-derived cells of the line PLC/PRF5/Alexander cells were obtained from the American Type Culture Collection (ATCC-No: CRL-8024).

### 2.6. Immunohistochemistry

Immunohistochemistry (IHC) was performed on FFPE specimens cut at 2–4 µm thickness as previously described [34]. Subsequently, slides were digitalized by a whole slide scanner at 40×, with a pixel size of 0.2278 × 0.2278 µm (Nanozoomer, Hamamatsu Photonics, Hamamatsu, Japan). IHC stains for E- and N-cadherin were manually evaluated by scoring the intensity of staining from 0 (no staining) to 3 (strong staining), as described before [37]. The mean value was calculated for patients with the same entity. Samples with a mean manual score of greater than or equal to 1.5 were assigned to the high group for E- or N-cadherin, while cases that did not reach this value were assigned to the low group. A quantitative assessment of E- and N-cadherin expression levels was performed using QuPath, an open-source bioimage analysis software (Bankhead; https://qupath.github.io/, accessed on 7 July 2022; version 0.2.3) [38]. Tumor cells were annotated using a detection classifier. The H-score was calculated from the extent and intensity of staining, giving a score range of 0 to 300. The antibodies used are listed in Appendix A. Double-labeling with E- or N-cadherin together with CD34 was performed with a Vector Red detection system (EnVision Flex HRP Magenta Chromogen, Dako Deutschland GmbH, Hamburg, Germany) and with 3,3′-Diaminobenzidine (EnVision Flex DAB Chromogen, Dako Deutschland GmbH, Hamburg, Germany). For evaluation of α-1-fetoprotein (AFP), glypican-3 (GPC3), and zinc finger E-box binding homeobox 1 (ZEB1), immunohistochemistry was performed and the immune reactive score (IRS) was assessed [39]. The IRS gives a range of 0–12 as a product of multiplication between positive cells proportion score (0–4) and staining intensity score (0–3). HCCs with a mean IRS of ≥5 were assigned to the high group.

### 2.7. Statistical Analysis

We performed statistical analyses within the R environment for statistical computing (version 4.1.2, R Foundation for Statistical Computing, Vienna, Austria) [40]. We compared differences between two independent groups when dependent variables were either ordinal or continuous with the non-parametric Mann-Whitney U test. The Kruskal–Wallis test was applied to compare two independent groups, which consist of one dependent scale variable and one explanatory nominal variable with three or more levels. We applied Benjamini-Hochberg corrections to reduce the effects of multiple testing and control for the false discovery rate. Spearman’s correlation was used to examine linear correlations between two numeric variables showing a non-normal distribution. *p* values ≤ 0.05 were considered statistically significant. E- and N-cadherin protein expression scores were dichotomized utilizing the R maxstat package to provide a significant distinction between the high and low expression levels based on survival outcome [41]. We defined overall survival as the interval between initial diagnosis and death, regardless of etiology or the last follow-up. Overall survival was calculated by the Kaplan–Meyer method. Differences were evaluated by the log-rank test. Multivariate analyses were performed using SPSS 27 Software (SPSS, Chicago, IL, USA). We retrieved TCGA expression data from http://www.oncolnc.org/ and https://xena.ucsc.edu/ (both accessed on 7 July 2022) and evaluated them as mentioned above [42,43]. The forest plot was created using Microsoft Excel 2016 (Microsoft Corporation, Redmond, WA, USA).

## 3. Results

### 3.1. E- and N-cadherin-heterodimers Are Retained in Different Species and during Hepatocyte Carcinogenesis

Initially, expression of the adherens junction transmembrane glycoproteins E- and N-cadherin was believed to be mutually exclusive within epithelial (E-cadherin) versus mesenchymal and neuroectodermal cells (N-cadherin); however, we previously discovered a novel AJ type on the basolateral membrane of hepatocytes in the normal liver, in which E- and N-cadherin colocalize as cis-E:N-cadherin heterodimers [22]. E- and N-cadherin-containing AJ are conserved across hepatocytes from different species (Figure 1) as demonstrated in cultured primary mouse and human hepatocytes (see Appendix B, Figure A1) via double-label laser-scanning fluorescence microscopy. Besides the colocalization of E- and N-cadherin in hepatocytes of the normal liver, we noted N-cadherin-positive, E-cadherin negative dot-like staining in sinusoids, presumably corresponding to the AJ of the mesenchymal hepatic stellate cells in the space of Disse [44].

To investigate whether E- and N-cadherin heterodimer-containing AJ are also retained in hepatocellular tumors and derived cell cultures, we performed immunoprecipitation experiments with antibodies against E- and N-cadherin and an unrelated antibody of the same species (mouse myeloma antibody) in whole-cell lysates of PLC cells, as well as in lysates of human normal liver and human HCC tissue. In PLC cells, using different detergent-soluble fractions, E-cadherin coprecipitated with N-cadherin, the plaque proteins α-and β-catenin, as well as plakoglobin, while N-cadherin was demonstrated in a complex with E-cadherin, α-, β-catenin, and vinculin (Figure 2). Interestingly, E- and N-cadherin complexes were especially enriched when using strong detergents (RIPA-buffer), which may help with solubilizing lipophilic membrane fractions. In contrast, we found an enrichment of cadherin–catenin complexes, as with α-catenin, using mild detergents (Triton-X-100). It is of note, however, that E-cadherin, N-cadherin as well as the plaque proteins α- and β-catenin coprecipitated in all buffers, albeit at different amounts. Immunoprecipitation experiments were also performed with other buffers, such as Empi-gen-containing immunoprecipitation buffers, but the recovery of E- and N-cadherin, as well as their protein complexes, was less. In whole-cell lysates of HCC tissue, E-cadherin coprecipitated with N-cadherin, and N-cadherin with E-cadherin (Appendix B, Figure A2, again using RIPA buffer), which demonstrates the in situ prevalence of E:N-cadherin heterodimers. Each time, in an immunoprecipitation experiment with an unrelated mouse myeloma antibody performed in parallel, no complexes with cadherins and catenins were obtained in the respective control immunoprecipitation experiments nor were cadherin–catenin complexes precipitated with non-antibody-bound beads precleared with the tissue or cell lysates. In corresponding HCC tissues, we observed complete colocalization of E- and N-cadherin using confocal laser scanning microscopy (Appendix B, Figure A2). To conclude, we could again demonstrate E:N-cadherin complexes in liver and derived hepatocellular carcinoma in situ as well as in cell culture. Concerning PLC cells (compare Figure 2), our data suggest complexes of E- and N-cadherin besides complexes of homodimeric E- and N-cadherin complexes with the respective catenins. Correspondingly, double label immunofluorescence microscopy in PLC cells showed membrane areas with complete colocalization of E- and N-cadherin besides areas of solely E- or N-cadherin positivity.

Therefore, to investigate whether E- and N-cadherin were detected in one complex in liver and HCC tissue in situ, we performed confocal laser scanning fluorescence microscopy of normal liver tissue, and both E- and N-cadherin were colocalized with the plaque proteins α- and β-catenin at the AJs at the border to the bile canaliculi and at the basolateral cell membrane (Figure 3). In contrast to E-cadherin, N-cadherin was localized together with α- and β-catenin in dot-like AJ, mutually corresponding to AJs of hepatic stellate cells (HSCs), whereas only catenins together with VE-cadherin lined the delicate AJs of liver sinusoidal endothelial cells (LSEC, see Géraud et al. [45]). In HCA and HCC in situ, colocalization of E- and N-cadherin with the respective catenins were retained (Figure 3), pointing to robust and stable complexes during hepatocarcinogenesis. In examples of well, moderately, and poorly differentiated human HCCs, E- and N-cadherin were colocalized together with the plaque proteins α and β-catenin (Figure 3), plakoglobin, and protein p120-ctn, but as expected not with the tight junction protein ZO-1; however, our analyses also visualized the distorted architecture of HCC with respect to apico-basolateral polarity as demonstrated by the tight junction protein ZO-1 in HCC cells, the increase in atypical vessels positive for protein ZO1 and p120 ctn, and the previously shown loss of N-cadherin-positive HSCs in HCC (Figure 3 and Figure 4).

As expected in malignant tumors, staining with the respective proteins of AJ and tight junctions demonstrated architectural disarray and loss of polarity (Figure 4, for comparison: Figure 1) [46]. Remarkably, even in poorly differentiated HCC, E- and N-cadherin were still colocalized to a large degree. In cells of the line PLC/PRF-5/Alexander, besides complete colocalization, also mutually exclusive staining pattern of E- and N-cadherin was observed in subconfluently grown cells (Appendix B, Figure A1), whereas complete colocalization was noted in confluently grown PLC cells. We have thus shown that both E- and N-cadherin are constitutively expressed in hepatocytes of different species, as well as in HCC and derived cell cultures, together with the respective plaque proteins. 

### 3.2. E- and N-cadherin Are Stably Expressed in Human Hepatocellular Tumors

Since we demonstrated stable E- and N-cadherin expression in normal liver, HCA, and HCC using protein biochemistry and fluorescence microscopy, we aimed to validate E-N-cadherin coexpression as a hallmark of cells of hepatocytic lineage in hepatocellular tumors; therefore, we used immunohistochemistry against E- and N-cadherin in tissue microarrays of a large cohort of 868 HCCs of 570 patients with respective non-neoplastic liver parenchyma mostly with liver cirrhosis, which also included dysplastic foci and dysplastic nodules as precursor lesions, multifocal HCC lesions, HCC recurrences, HCC lymph node, and foreign metastases as well as vascular invasions. In addition, benign and non-neoplastic liver tumors were analyzed, including 25 HCA and 31 FNH with corresponding non-neoplastic liver tissue, respectively. E- and N-cadherin was analyzed and scored manually from negative (score 0) to strongly positive (score 3) as well as digitally with the software program QuPath. E-cadherin was the most stable immunohistochemical marker, with consistently over 90% of patient samples assigned to the high expression group, while N-cadherin staining was slightly weaker with minor fluctuations with a proportion of over 80% (Table 1, Figure 5). Minor fluctuation of marker expression as detected by immunohistochemistry may also be attributed to different fixation statuses after surgical removal. In line with this, also normal liver parenchyma showed slight variations in E- and N-cadherin staining; however, in our analysis, these fluctuations were independent of whether low- or high-grade HCC or HCC metastases were investigated. 

By immunohistochemistry, membranous E- and N-cadherin staining was detected at nearly equal amounts in hepatocytes of the normal and cirrhotic liver, FNH, DN, HCA, and HCC (Figure 5), while reduced E- as well as N-cadherin staining was only observed in few higher-grade HCC metastases. As expected, there was an overall correlation between E- and N-cadherin expression in both manual and digital immunohistochemical evaluation (*p* < 0.01, *ρ* = 0.36 and *ρ* = 0.38). Between manual and digital assessment, E-cadherin (*p* < 0.01, *ρ* = 0.82) as well as N-cadherin (*p* < 0.01, *ρ* = 0.82) showed a strong positive mutual relationship. These results indicate that E- and N-cadherin are robustly expressed during hepatocarcinogenesis, which does not fundamentally change in the context of dedifferentiation or metastasis.

### 3.3. N-cadherin Is Not a Suitable Marker for EMT in HCC

Downregulation of E-cadherin and upregulation of N-cadherin are viewed as hallmarks of EMT in diverse tumor entities [47,48]. Normal hepatocytes harbor nearly equal amounts of E- and N-cadherin, which is maintained during hepatocarcinogenesis. In addition, E- and N-cadherin are downregulated in high-grade HCC. This may suggest a function distinct from the common concept of EMT. Driven by this hypothesis, we correlated our semiquantitative E- and N-cadherin immunohistochemical scores with known prognostic markers in HCC, which also included additional markers of EMT.

For this purpose, we selected HCC cases from our cohort with previously described vessels encapsulating tumor clusters (VETC)-pattern, which is known to confer unfavorable prognosis [49], and performed double-label immunohistochemistry against CD34 and E- or N-cadherin (Figure 6). Tumors showing a VETC pattern showed, on average, significantly lower E-cadherin and significantly higher N-cadherin expression, although E- and N-cadherin amount was rather similar. No significant alterations of E- or N-cadherin were observed in vascular invasions or the invasive HCC border. 

### 3.4. E-cadherin Expression Is Not an Independent Prognostic Marker for Overall Patient Survival in HCC

In addition to HCC with VETC-pattern, we investigated other prognostic markers for correlations with the immunohistochemical expression of E- and N-cadherin. While a significant difference in overall survival was observed after stratification of E-cadherin expression (Figure 7A), no difference was observed for N-cadherin (Figure 7B). With the cutoff finder method, a cutoff of an E-cadherin value of 2.57 was determined. Performing subgroup analysis, we identified vascular invasion as a possible confounder: patients with low E-cadherin showed significantly more vascular invasions, which are associated with poorer survival. AFP, which is a valuable in situ and serum marker for the hepatocyte lineage, and also the most important prognostic marker in HCC, showed no significant difference in immunohistochemistry (Figure 7C) or serum levels (r_s_ (E-cadherin) = 0.06, *p* = 0.42, *n* = 188; r_s_ (N-cadherin) = −0.04, *p* = 0.641, *n* = 188). Similarly, we did not find significant differences when comparing both groups in terms of GPC3 expression. Interestingly, ZEB1, a frequently analyzed and well-described marker for EMT was neither associated with the expression of E-cadherin nor N-cadherin. In addition, no significant association was detected with the HCC subtype, such as macrotrabecular-massive HCC-subtype, tumor stage or tumor grade. Multivariate Cox regression analysis was performed to evaluate the prognostic relationship between E-cadherin, N-cadherin, vascular invasions, as well as AFP, GPC3, and ZEB1 immunohistochemistry. AFP staining and vascular invasions showed statistically significant results, but GPC3 did not. The hazard ratios and 95% confidence intervals were 1.11 and 1.05–1.18 for AFP, and 1.78 and 1.42–2.24 for vascular invasions (each *p* < 0.001). In this model, E- and N-cadherin were not independent prognostic factors, supporting the hypothesis of vascular invasions as a confounder in the survival analysis. 

In summary, E- and N-cadherin alone are no good markers for the prediction of overall survival. The separation of overall survival between patients with high and low E-cadherin expression is primarily due to minimally reduced E-cadherin expression in patients with vascular invasion; however, the expression of E-cadherin on protein level as investigated in our immunohistochemistry cohort is mostly still preserved, so it should not be interpreted as changes in the context of a possible EMT. Furthermore, there is no association between E-and N-cadherin and AFP serum levels, AFP immunohistochemistry, prognostic factors such as GPC3, or EMT-associated factors such as ZEB1. Consecutively, in our large patient cohort, E- and N-cadherin are not independent prognostic factors. Furthermore, neither E-cadherin nor N-cadherin expression are associated with EMT.

### 3.5. E- and N-cadherin Do Not Serve as Direct Predictors of Overall Survival in HCC in the Independent TCGA Cohort

Finally, we intended to validate our results in an independent cohort on the mRNA level. Applying the TCGA cohort, we found that low mRNA levels of *CDH1* were associated with an unfavorable prognosis, whereas no significant effect on survival was observed in the analysis of *CDH2* (Appendix C, Figure A3), which is in line with immunohistochemical analyses in our large cohort. Regarding mRNAs of the AJs plaque components, high mRNA levels of *CTNNA1* and *VCL* correlated with poor survival, and high mRNA levels of *JUP* and the tight junction component *TJP1* with improved survival, whereas mRNA levels of *CTNNB1* and *CTNND1* did not show any clear trend. To further elucidate a potential relationship between E- or N-cadherin expression and the survival of HCC patients, we performed a Cox proportional hazards regression of mRNA levels. For this purpose, we analyzed structural proteins (*CDH1*, *CDH2*, *KRT19*, *LAMA3*, *LAMC2*, and *MMP9*) and transcription factors (*CD151*, *ID2*, *SNAI1*, *SNAI2*, *TCF3*, *TGFB1*, *TWIST1*, and *ZEB1*) already described as prognostic markers in HCC [50]. In addition to their prognostic properties, all of these factors have been described to be associated with EMT. Here, in contrast to our survival analysis, *CDH1* was not shown to be significantly altered (Figure 8). In our model, *CD151*, *TCF3*, and *ZEB1* had the greatest impact on overall survival.

## 4. Discussion

Here, we demonstrate in a large cohort of HCC and HCA patients, as well as respective non-neoplastic/normal livers, that E- and N-cadherin expression constitute a general characteristic of hepatocellular differentiation, and may therefore be used as a diagnostic marker for hepatocellular/liver origin; however, our data do not suggest E- and N-cadherin as suitable markers for EMT in non-neoplastic and neoplastic liver tissue.

### 4.1. E- and N-cadherin Are Constitutively Expressed in AJs of Hepatocytes and Derived Tumors

While E- and N-cadherin have been extensively studied for their use as markers in EMT, there are only a few comparative protein biochemical studies investigating AJs in non-neoplastic and neoplastic liver tissue. Our data obtained from immunofluorescence microscopy and protein biochemistry using immunoprecipitation showed that E- and N-cadherin completely colocalize at the cell membrane in a complex together with plaque proteins α- and β-catenin, plakoglobin, and vinculin in normal hepatocytes, HCA and HCC in situ. In slight contrast, in cultured cells of the line PLC PRF-5/Alexander, besides E:N-cadherin heterodimeric complexes, also E- and/or N-cadherin homodimeric complexes were detected and also different catenin subgroups found in immunoprecipitation experiments and verified in double label immunofluorescence microscopy. The parallel existence of E:N-cadherin, as well as E-cadherin and N-cadherin complexes in PLC cells, were matched by the presence of heterodimeric junctions in normal human hepatocytes in situ, which also contained E:N-cadherin heterodimeric complexes besides heterophilic N-cadherin junctions towards mesenchymal liver cells such as hepatic stellate cells, a phenomenon requiring further investigation in the future; thus, we extended our study concerning cis-E:N-cadherin heterodimers [22] in endoderm-derived cells. To exclude any bias, we relatively quantified the immunohistochemical scores for E- and N-cadherin both manually (E- and N-cadherin score) and via computer-assisted bioimage analysis (E- and N-cadherin H-score). While both scores consistently showed correlations, however, the computer-assisted analysis was less reproducible, so we chose the manual evaluations for further analyses. The values were dichotomized using a low- and a high-score group. The low score group included all tissues that showed very weak to weak expression, while the high score group included all tissues that showed at least moderately strong expression. By immunohistochemistry, we thereby demonstrated that both E- and N-cadherin are expressed in non-neoplastic liver tissue as well as in HCA, HCC, foreign and lymph node metastases, and sites of vascular invasion. Rare cases of sarcomatoid/dedifferentiated HCC were positive only for N-cadherin and not E-cadherin, which is suitable for the notion of loss of hepatocytic differentiation. In contrast, poorly differentiated HCC with at least in part retained hepatocytic differentiation was still characterized by coexpression of E- and N-cadherin, albeit at overall reduced expression levels. Minor variations in expression patterns may also be due to changes in the growth pattern, which is reflected by HCC subtypes and also play a prognostic role in HCC [51]. Similar to E-cadherin, tight junctions are pivotal in retaining hepatocyte polarity [52]. In the TCGA cohort, HCCs with low *TJP1* mRNA levels coding for the tight junction protein ZO-1 had significantly worse survival. This is consistent with a study by Yokota et al. showing that exosomal miRNA of a highly metastatic cancer cell line may affect vascular permeability by downregulation of ZO-1, and that this is a negative prognostic marker in HCC [53]. Our data show the downregulation of AJ components in the context of hepatocarcinogenesis. On the mRNA level in the TCGA cohort, this seems not to apply to catenins); however, it is of note that especially β-catenin has a dual function in AJ and Wnt-signaling cascade and that *CTNNB1* is frequently mutated in HCC and some HCA, which leads to Wnt-activation [54,55]. One of the most important mutations within the Wnt/β-catenin pathway are mutations of the adenomatous polyposis coli (APC) gene in colorectal carcinoma [56], which influence downstream β-catenin [56,57]; however, APC mutations play a subordinate role in HCC [58]. Activating mutations of *CTNNB1* occur in up to 30% of HCC, while APC mutations affect only 1-3% of HCC, and both alterations are mutually exclusive [14]. In mice, a homozygous *CTNNB1* mutation leads to embryonic lethality [59]. This is also described for homozygous deletion of *CDH1* and *CDH2* and other AJ plaque components, thus highlighting the overall importance of AJ for normal cell and tissue homeostasis. In HCC, mutations in *CTNNB1* affect signal transduction; however, loss of β-catenin is compensated by the upregulation of plakoglobin, resulting in retained structural properties of AJs [60,61]. Overall, structural proteins of AJs, including β-catenin and E-cadherin, appear to be critical to maintain cellular integrity.

### 4.2. E- and N-cadherin in the Context of EMT

Given the consistent expression of E- and N-cadherin in immunohistochemistry and immunofluorescence microscopy in normal hepatocytes, HCA and HCC, a direct role in classic EMT, which describes downregulation of E-cadherin as well as de novo expression of N-cadherin during carcinogenesis, tumor invasion, and metastasis, does not hold true for HCC. Thus, we postulate that N-cadherin is not a suitable derivative marker for EMT in HCC. While previous work characterized E-cadherin, N-cadherin as well as the expression of respective catenins in the context of hepatocarcinogenesis, these studies used significantly smaller cohorts of patients [62,63,64]. We found that E- and N-cadherin are both decreased during hepatocarcinogenesis, and the highest amounts were seen in non-neoplastic livers. This is in contrast to previously published studies that reported E-cadherin to be exclusively expressed in epithelia, and N-cadherin in normal neural/neuroectodermal and mesenchymal cell types. Furthermore, E-cadherin expression was shown to decrease from HCA to HCC in humans [65] and mice [66]. Along the same line, another study demonstrated that E-cadherin was immunohistochemically detected in hepatocytes, HCA, well and moderately differentiated HCC, and only lost in poorly or dedifferentiated HCC [67]. To our knowledge, N-cadherin has not yet been analyzed comprehensively in HCA. In our cohort, we did not find significant correlations between deregulated N- or E-cadherin and HCA subtypes or BHCA with known *CTNNB1*-mutation. This finding supports our assumption that E- and N-cadherin are regularly expressed in hepatocytes and hepatocytic tumors. Otherwise, one may expect a gradual decrease in E-cadherin expression from normal tissue to HCA to HCC in the context of EMT. Correspondingly, an increase in N-cadherin would be expected, which we did not find in our large cohort, nor was it detected locally in invasive tumor areas or HCC hemangioinvasion. According to current knowledge, EMT is discussed to be an extremely complex process, so not all criteria of EMT may be fulfilled. Liang et al. were unable to show a singular generic EMT-related gene signature but described five different patterns instead. In their 773-gene list of potential candidate genes, they identified several pathways of EMT, of which only one cluster was associated with disruptions in cell junctions. Other critical groups of genes whose alteration may enable EMT include genes involved in cell growth, migration, apoptosis, transcriptional regulation, and stem cell maintenance [68]. Furthermore, the biological roles of E- and N-cadherin are versatile, explaining the controversial roles of E- and N-cadherin in different tumors as already described by van Roy [69]. In numerous carcinomas, E-cadherin shows tumor-suppressing activities, while in inflammatory breast, ovarian, and squamous carcinomas, E-cadherin has been described to lead to tumor-promoting microembolus formation and prolonged survival via *EGFR* signaling [69]. Similar controversial roles have been described for N-cadherin. In neuroblastoma, low N-cadherin expression is correlated with metastasis [70], while in melanoma [71] and several other carcinomas [72], high levels of N-cadherin have been associated with the induction of genes involved in cancer progression and have been linked to EMT. Only a few studies have investigated the expression of N-cadherin in HCC with regards to clinicopathological data. While some authors described an association between reduced N-cadherin expression with poor disease-free survival [64,73], other authors associated increased N-cadherin expression with unfavorable patient outcomes [63,74]. In our large patient cohort, no significant association was observed using N-cadherin immunohistochemistry; however, reduced N-cadherin mRNA was associated with poor survival in the TCGA cohort, which is counterintuitive to the EMT hypothesis. Based in our findings, we consider high expression of both E- and N-cadherin together with the formation of heterodimers in hepatocytes and derived tumors as an integral part of the assembly of these cells. This is supported by the discordant role of E-cadherin in relation to N-cadherin, which implies that the EMT concept remains inconclusive with regards to both of these cadherins in HCC.

### 4.3. Prognostic Properties of E-cadherin in HCC Are an Epiphenomenon, While N-cadherin Does Not Predict Overall Survival in HCC Patients

A correct estimation of prognosis in cancer patients may improve clinical decision-making and consequently patient outcomes. Although several markers associated with EMT may serve as prognostic markers of survival in HCC, the prognostic properties should be considered isolated. In a meta-analysis by Chen et al., an overall reduction in E-cadherin expression was associated with an unfavorable prognosis in HCC; however, the percentage of HCC with reduced expression of E-cadherin in the selected studies had a variance between 1.6% and 76.2%. In addition, several analyses used E-cadherin as a marker for EMT to complement other features of malignancy, which led to the prognostic properties being considered merely as evidence of EMT. [75] Based on the properties of E- and N-cadherin in HCC, a direct impact of E- and N-cadherin on the prognosis of HCC patients seems rather unlikely.

To support this hypothesis, we additionally analyzed several prognostic factors in HCC with a focus on tumor vessel formation and hemangioinvasion, which represent features characteristic of progressed HCC with prognostically unfavorable outcomes. For this purpose, we have chosen to analyze E- and N-cadherin in VETC-positive HCC, showing intimate contact of HCC cells with vessels. In addition to its prognostic value, the VETC pattern is independent of EMT. Intriguingly, this revealed an association between VETC and both E- and N-cadherin. In VETC-positive HCCs, we observed slightly less E-cadherin and slightly more N-cadherin expression. In the original publication by Fang et al., fewer cases with reduced E-cadherin expression were found in the VETC-positive HCC compared to the VETC-negative HCC, while N-cadherin was not investigated [76]; however, given the minimal differences, the link between the VETC pattern and E-cadherin expression may be explained by the significantly higher statistical power in our larger cohort—Fang’s group did not only analyze E-cadherin, but also other factors of EMT, such as Snail, Slug, and Twist, which may validate the notion that there is no relationship between VETC and EMT. N-cadherin may play a role in the formation of heterotypic junctions between HCC cells and endothelia [45,77] in VETC–positive HCC, but also in frequent hemangioinvasions, which are commonly found in progressed HCC (see also [22]). As a limitation, however, it must be mentioned here that our analyses using human HCCs in situ cannot fully depict dynamic processes as described in the context of EMT [78]. Our own preliminary data suggest a role of N-cadherin in heterotypic junctions of hepatocytes with HSC, whereas both N- and E-cadherin, were absent from the AJ complexes of LSEC; however, this may not exclude the possibility of heterotypic junctions of N-cadherin with mesenchymal/endothelial cells (see also Straub et al., 2011). This constellation alone, if true, may reconcile the differences present in our study with the results from Fang’s group.

Numerous factors are known to determine the clinical course of HCC patients [79], including vascular invasions [80], AFP [36,81,82], GPC3, ZEB1 [83], and VETC [49,76]. Thus, we analyzed overall survival together with these prognostically relevant factors, but found no associations between E- and N-cadherins and AFP, GPC3, or ZEB1. 

Therefore, we examined several prognostic factors comparatively with E- and N-cadherin. These markers were also associated with EMT in HCC in addition to their predictive properties [50]. For example, a four-gene signature of E-cadherin (*CDH1*), inhibitor of DNA binding 2 (*ID2*), matrix metalloproteinase 9 (*MMP9*), and transcription factor 3 (*TCF3*) was described. This signature describes a group of HCC patients with increased invasiveness and increased metastases, thus predicting prognosis. Downregulation of *CDH1* and *ID2* is associated with a negative prognosis, as well as upregulation of *MMP9* and *TCF3*. [84] Overexpression of Snail (*SNAI1*), Twist (*TWIST1*), and Slug (*SNAI2*) has also been described to negatively predict survival and to be associated with invasiveness and metastasis in HCC [81]. High expression of *CD151* leads to amplification of the integrin α6β1-PI3K signal in HCC and may therefore increase the aggressiveness of tumor aggressiveness. Consequently, CD151 has been proposed as a therapeutic target in HCC [82]. The extracellular matrix protein laminin-332 (*LAMA3* and *LAMC2*; formerly known as laminin 5), cytokeratin 19 (*KRT19*), and transforming growth factor beta 1 (*TGFB1*) have also been associated with EMT and negative prediction of survival in HCC [85,86,87]. In our analysis, *CD151*, *TCF3*, and *ZEB1* were shown to be the strongest predictors of overall survival in the TCGA cohort, while genes associated with structure-associated proteins had only a small or insignificant impact; however, it should be noted that E12/E47, the gene products of *TCF3*, as well as Twist, the protein of *TWIST1*, are repressors of E-cadherin [87,88], so an association is a logical consequence of those effectors. This would also explain why *CDH1* is no longer significant in a Cox regression model when *TCF3* and *TWIST1* are simultaneously considered, but shows an association with survival when *CDH1* is analyzed in isolation. Nevertheless, it is remarkable that both E- and N-cadherin (*CDH1*, *CDH2*), which are considered key proteins of invasiveness and metastasis, and hence, EMT, have no significant prognostic effect. 

### 4.4. The Diagnostic Use of E- and N-cadherin as Markers for Tumors of Hepatocytic/Liver Origin

Having demonstrated the formation of E:N-cadherin heterodimers in the liver, a feature that is highly preserved during hepatocarcinogenesis, the presence of both cadherins may be used for the differential diagnoses of carcinomas of unknown primary. In line with this, we have previously shown the presence of E- and N-cadherin in cholangiocytes of the biliary tree and derived tumors, suggesting N-cadherin as a marker to distinguish between intrahepatic cholangiocarcinoma and liver metastases of ductal adenocarcinoma of the pancreas [89]. In conclusion, E- and N-cadherin are suitable markers to identify hepatocytes, cholangiocytes, and derived tumors. This distinct structural configuration distinguishes liver parenchyma and its derived tumors from other epithelial tumors for which, according to the EMT hypothesis, alteration of E- and N-cadherin during carcinogenesis is considered characteristic and which predominantly express no or very little N-cadherin. Thus, N-cadherin, together with E-cadherin positivity, may be used to differentiate primary liver carcinomas such as HCC and intrahepatic cholangiocarcinoma from liver metastases of extrahepatic primary tumors, such as other gastrointestinal adenocarcinomas positive only for E-cadherin, but not N-cadherin, and may therefore be useful in routine histopathologic diagnostics.

## 5. Conclusions

Our work aimed to comprehensively analyze E- and N-cadherin in hepatocytes and derived tumors for their use as stable markers of hepatocytic differentiation. We have previously described E-N-cadherin heterodimers in endoderm-derived cells, so we hypothesized that E- and N-cadherin might play a major role in liver and primary liver tumors, thereby distinct from other organs. When one considers the constantly high amount of E- and N-cadherin in hepatocytes of normal liver, HCA, and HCC, without significant association with prognostic markers, the role of E- and N-cadherin in EMT in hepatocarcinogenesis should be reconsidered. 

## Figures and Tables

**Figure 1 cells-11-02507-f001:**
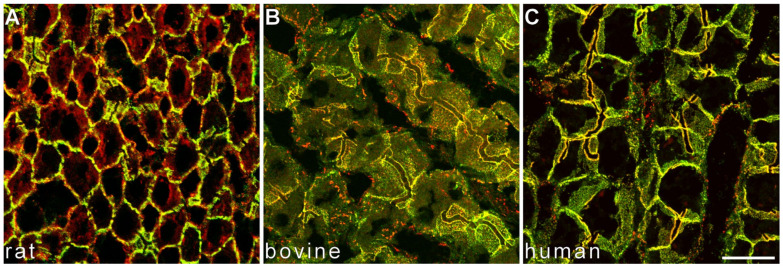
N-cadherin colocalizes with E-cadherin in hepatocytes of normal liver. Double label laser scanning microscopy shows near-complete colocalization of E-cadherin (**A**–**C**, green) and N-cadherin (**A**–**C**, red) in hepatocytes of rat (**A**), bovine (**B**), and human liver (**C**). Bar: 20 µm.

**Figure 2 cells-11-02507-f002:**
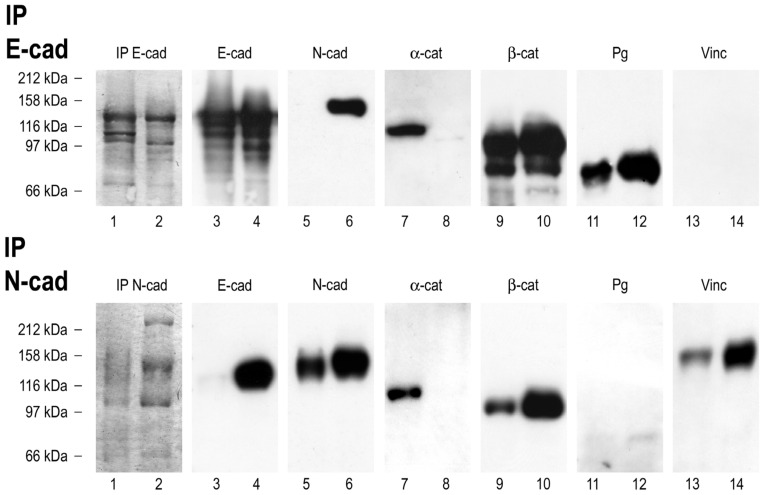
Identification of cis-E:N-cadherin heterodimers in PLC cell lysates, demonstrated by immunoprecipitation, SDS-PAGE (8%), and immunoblotting. Coomassie-Blue-stained major polypeptides of the immunoprecipitates obtained from 1% Triton X 100 soluble fractions (lane 1), as well as RIPA soluble fractions (lane 2) of PLC cells from protein G-bound antibodies against E-cadherin (above) and N-cadherin (below). Proteins of immunoprecipitates subjected to SDS-PAGE were probed by immunoblotting with antibodies against E-cadherin (lanes 3 and 4), N-cadherin (lanes 5 and 6), α-catenin (lanes 7 and 8), β-catenin (lanes 9 and 10), and plakoglobin (lanes 11 and 12). In each case, on the left hand, immunoprecipitates from Triton X 100 soluble lysates, and on the right hand, immunoprecipitates from RIPA-soluble lysates are shown. Note enhanced coimmunoprecipitation of E- and N-cadherin using RIPA-buffer (IP N-cadherin, lane 4 and IP E-cadherin, lane 6, respectively). On the contrary, α-catenin was coprecipitated with N-cadherin to a greater amount using 1% Triton X 100. Note faint band also detectable using RIPA -soluble lysates. Plakoglobin, however, seems to be physically associated rather with E-cadherin than with the N-cadherin complex. Molecular weight markers are indicated on the left.

**Figure 3 cells-11-02507-f003:**
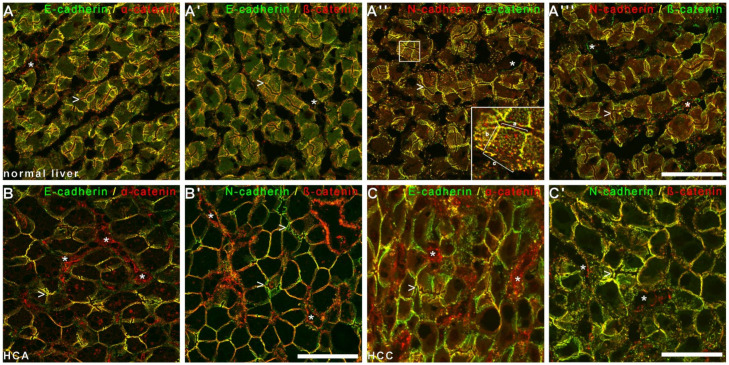
Colocalization of E- and N-cadherin with α- and β-catenin in normal liver, HCA, and HCC. Confocal laser scanning immunofluorescence microscopy of normal bovine liver (**A**–**A**‴), as well as human HCA (**B**,**B′**) and human HCC (**C**,**C′**) with antibodies against E-cadherin (mouse, Alexa 488, green), and N-cadherin (**A**″,**A****‴**: rb, cy3, red, **B′** and **C′**: mouse Alexa 488, green) together with antibodies against α- and β-catenin (**A**,**A′**, **B**,**B′**, **C**,**C′**: rb cy3, red; and **A****″**,**A****‴**: mouse Alexa 488, green) showed complete colocalization at the membranes of hepatocytes and hepatocellular tumor cells. Vascular spaces are lined with endothelial cells that stain with α- and β-catenin, but not E- and N-cadherin (stars). For a frontal plane of hepatocytes see insert in **A****″** with AJs of the canaliculi in direction of the apical membrane (a, see also arrowheads in all images), lateral membrane (b), and basal membrane (c). At the basal membrane, dot-like AJ complexes of N-cadherin together with α- and β-catenin, but not E-cadherin are noted, mutually corresponding to AJs of hepatic stellate cells (**A****″**, c, compare **A**,**A′** to **A****″**,**A‴**). Bars: each 50 µm.

**Figure 4 cells-11-02507-f004:**
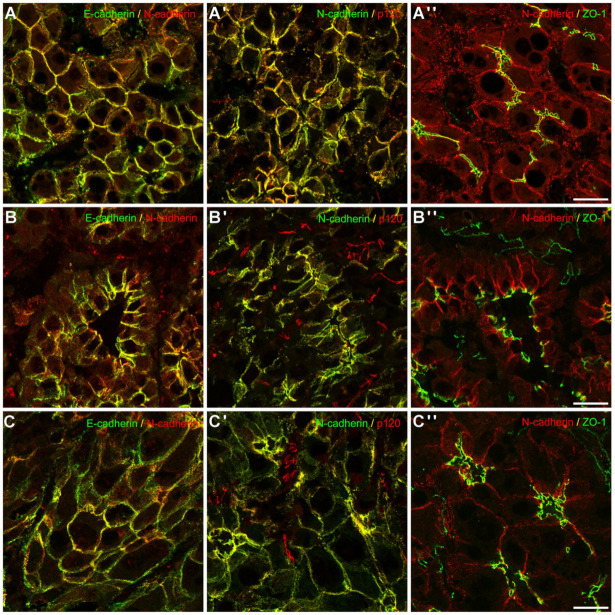
Immunofluorescence microscopy of components of junction components in grades 1, 2, and 3 HCC. E- and N-cadherin are colocalized in well (**A**), moderately (**B**), and poorly (**C**) differentiated HCC, together with the plaque protein p120-ctn (**A′**–**C′**). In normal liver tissue, tight junction protein ZO-1 is an indispensable component of the hepatic barrier and is required for apical polarity. In HCC, protein ZO-1 demonstrates the disruption of the cell polarity (**A″**–**C″**). Note also staining of N-cadherin as well as p120-ctn and protein ZO-1 in the small vessels. Bar: 20 µm.

**Figure 5 cells-11-02507-f005:**
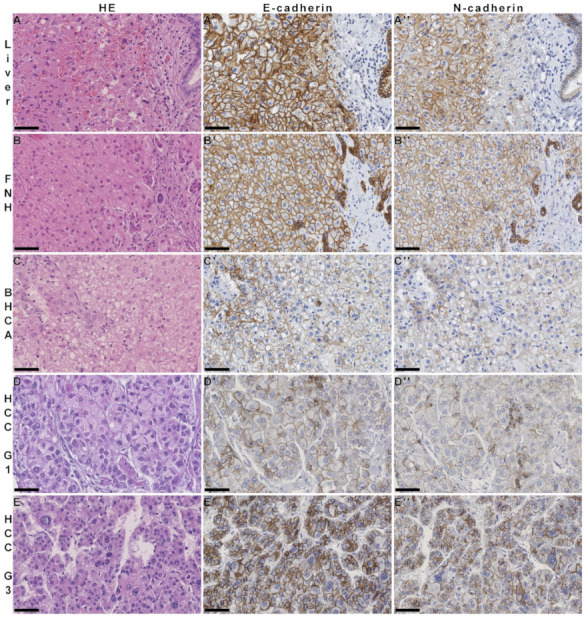
Immunohistochemistry of E- and N-cadherin in non-neoplastic liver and derived tumors. In each row, hematoxylin and eosin staining as well as E-cadherin (**A′**–**E′**) and N-cadherin (**A″**–**E″**) immunohistochemistry of normal liver tissue (**A**), FNH (**B**), β-catenin-activated HCA (BHCA; (**C**)), well-differentiated HCC (HCC G1; (**D**)) and poorly differentiated HCC (HCC G3, (**E**)). E- and N-cadherin are both coexpressed in hepatocytes, HCA, and HCC. Note slightly zonal expression pattern in normal liver with lower N-cadherin expression and higher E-cadherin expression in normal bile ducts (**A**–**A″**). Bars: 50 µm each.

**Figure 6 cells-11-02507-f006:**
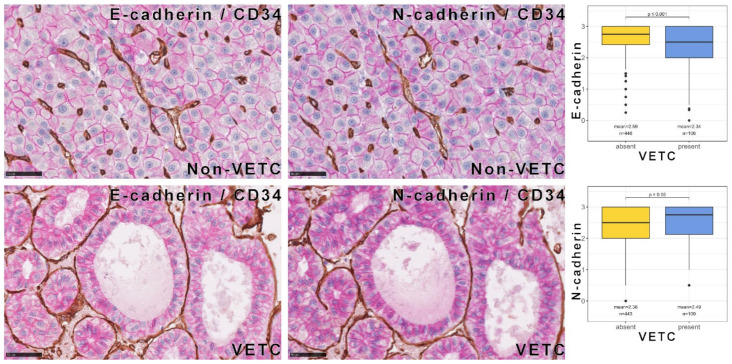
Association between E- and N-cadherin expression and VETC pattern in HCC. Double-labeling immunohistochemistry of CD34-expressing vessels (brown) with E- and N-cadherin (red). The differences are very small in individual cases and are only evident in the statistical analysis of the very large cohort (right panel).

**Figure 7 cells-11-02507-f007:**
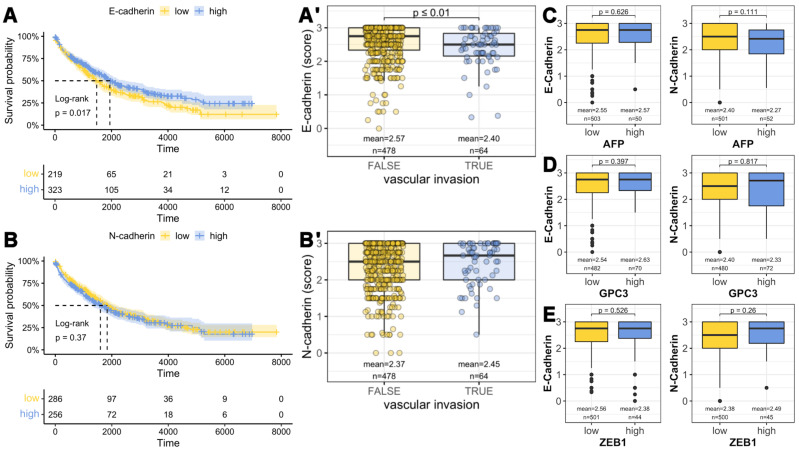
Kaplan–Meier overall survival in our cohort and E-/N-cadherin expression in relation to prognostic factors. Stratification according to E-and N-cadherin levels ((**A**), E-cadherin; (**B**), N-cadherin). Time is depicted in days. Differences in survival stratified by E-cadherin levels may be linked to significantly lower E-cadherin in HCC patients with vascular invasion (**A′**), while for N-cadherin, no significant association was detected (**B′**). With immunohistochemistry, no association of prognostically relevant proteins such as AFP (**C**) and GPC3 (**D**) ZEB1 (**E**), a prognostic marker associated with EMT, was found with E- or N-cadherin.

**Figure 8 cells-11-02507-f008:**
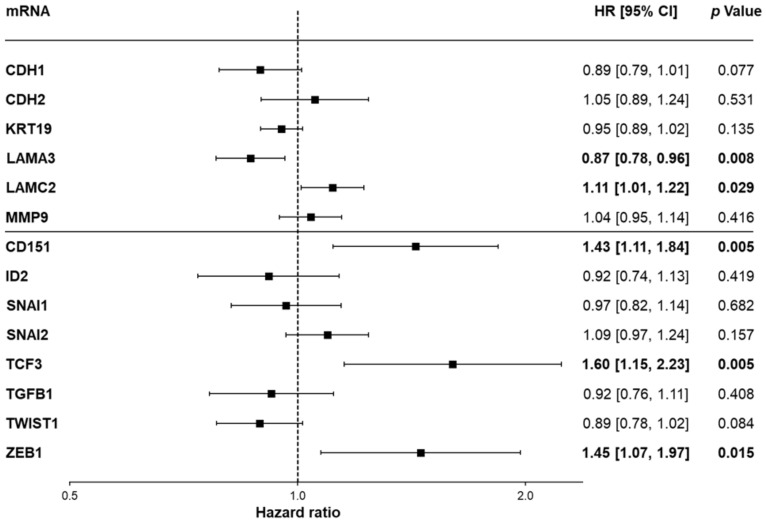
Cox regression analysis of mRNA levels of prognostic structural proteins and transcription factors in the TCGA cohort. Forest plot depicting the relationship between the overall survival and the mRNA expression of multiple genes. Of note, the structural proteins (upper half) such as *CDH1* and *CDH2* have only low prognostic value in HCC, in contrast to the transcription factors such as *CD151*, *TCF3*, and *ZEB1* (lower half). HR: hazard ratio. 95% CI: 95% confidence interval.Significant results are highlighted in bold.

**Table 1 cells-11-02507-t001:** Immunohistochemical evaluation of E- and N-cadherin as well as Ki-67 in HCC, HCA, FNH, and non-neoplastic liver tissue.

			E-cadherin			N-cadherin			
Category	Subcategory	Cases	Score ^†^	H-Score ^†^	High [%]	Score ^†^	H-Score ^†^	High [%]	Ki-67 ^†^ [%]
HCC	Metastases	72	2.41 ± 0.72	123.04 ± 64.7	92.42	2.09 ± 0.85	80.96 ± 55.33	85.29	10.28 ± 13.08
	Vascular invasion	32	2.59 ± 0.56	137.77 ± 59.75	96.67	2.32 ± 0.64	93.12 ± 47.49	93.55	13.71 ± 12.92
	High grade	94	2.48 ± 0.62	121.59 ± 49.89	95.56	2.3 ± 0.69	94.13 ± 46.98	90	14.68 ± 14.72
	Intermediate grade	396	2.55 ± 0.54	126.8 ± 46.61	96.67	2.41 ± 0.62	101.51 ± 46.63	93.57	7.33 ± 8.86
	Low grade	90	2.56 ± 0.55	134.26 ± 47.94	97.73	2.35 ± 0.66	102.02 ± 47.76	89.53	6.65 ± 8.63
	HCC from HCA	3	2.17 ± 0.29	56.65 ± 28.64	100	1.83 ± 1.04	42.24 ± 44.11	66.67	n.a.
	Total patients	570	2.55 ± 0.55	125.94 ± 46.48	96.4	2.38 ± 0.64	100.28 ± 47.3	92.64	8.73 ± 10.65
HCA	BHCA/BIHCA	5	2.5 ± 0.47	100.6 ± 61.51	100	2.15 ± 1.02	43.09 ± 34.49	80	n.a.
	IHCA/HHCA/UHCA	20	2.75 ± 0.35	109.84 ± 56.73	100	1.93 ± 0.74	38.19 ± 30.02	80	n.a.
	Total patients	25	2.7 ± 0.38	107.99 ± 56.5	100	1.97 ± 0.78	39.17 ± 30.26	80	n.a.
DN		12	2.66 ± 0.42	142.59 ± 53.83	100	2.34 ± 0.56	110.56 ± 56.95	100	2.36 ± 2.23
FNH		31	2.58 ± 0.45	96.12 ± 59.21	100	2.06 ± 0.65	60.83 ± 50.65	87.1	1.33 ± 1.1
Non-neoplastic Liver	Non-cirrhotic	281	2.59 ± 0.47	123.02 ± 44.08	99.29	2.21 ± 0.58	89.92 ± 41.84	94.46	3.32 ± 4.72
	Cirrhotic	359	2.51 ± 0.48	123.01 ± 37.94	98.24	2.11 ± 0.61	89.63 ± 39.25	90.64	8.25 ± 9.8
	Total patients	640	2.54 ± 0.48	123.01 ± 40.86	93.71	2.15 ± 0.59	89.76 ± 40.45	92.33	6.24 ± 8.49

^†^ Mean value ± standard deviation. n.a.: not analyzed/data not available.

## Data Availability

The datasets generated during and/or analyzed during the current study are available from the corresponding author on reasonable request.

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
