# Peer review of "Constitutive Occurrence of E:N-cadherin Heterodimers in Adherens Junctions of Hepatocytes and Derived Tumors"

_cells, 2022, doi:10.3390/cells11162507_

Round 1

Reviewer 1 Report

I read with interest the paper by Gerber and collegues. The authors analyzed the expression of E- and N- cadherin in a large series of hepatocellular lesions and normal hepatocytes and proved that E/N ratio is preserved along different steps of human hepatocarcinogeneisis, including high grade HCC and metastases. They concluded that the co-expression of E/N is a general characteristic of  hepatocyets; at the same time this finding aloow to conclude that HCC are not characterized by epithelial to mesenchymal transition.

Then the researchers investigated the correlation of E/N with clinical parameters and markers of angioinvasion. Their results showed that E/N is not correlated with prognosis or prognostically relevant factors. On the other hand, they found that the recently described vascular phenotype VETC is associated with E cadherin reduction and N-cadherin increase and discussed this finding.

The paper has a robust scientific approach and a certain degree of originality. However, there are some aspects that need to be clarified.

Major aspects.

  1. The authors should state clearly what is the aim of the study (description of E/N distribution in hepatocellular lesion? To investigate the prognostic role of E/N? Others?)
  2. The results regarding EMT and prognosis should be separated in separated sections
  3. The results regarding the prognosis should be explained better: was any multivariable analysis performed? Why vascular invasion is a confounder? Is E cadherin associated or not with prognosis?
  4. The discussion regarding prognosis and prognostic factor should be reformulated in a more clearly, less repetitive and more incisive way (taking benefit of aims of the study).

Author Response

I read with interest the paper by Gerber and collegues. The authors analyzed the expression of E- and N- cadherin in a large series of hepatocellular lesions and normal hepatocytes and proved that E/N ratio is preserved along different steps of human hepatocarcinogeneisis, including high grade HCC and metastases. They concluded that the co-expression of E/N is a general characteristic of hepatocyets; at the same time this finding aloow to conclude that HCC are not characterized by epithelial to mesenchymal transition.

Then the researchers investigated the correlation of E/N with clinical parameters and markers of angioinvasion. Their results showed that E/N is not correlated with prognosis or prognostically relevant factors. On the other hand, they found that the recently described vascular phenotype VETC is associated with E cadherin reduction and N-cadherin increase and discussed this finding.

The paper has a robust scientific approach and a certain degree of originality. However, there are some aspects that need to be clarified.

Major aspects.

  1. The authors should state clearly what is the aim of the study (description of E/N distribution in hepatocellular lesion? To investigate the prognostic role of E/N? Others?)

Response: Thank you very much for your evaluation of our manuscript and your valuable comments! We have reformulated the objectives of our study more clearly and refer to these objectives in the discussion. Our major aim is the description of E/N-cadherin as markers for hepatocytic differentiation. Yet secondary, the occurrence of E-/N-cadherin heterodimers affects also the popular concept of EMT in carcinogenesis.

  1. The results regarding EMT and prognosis should be separated in separated sections.

Response: We have subdivided the two sections in the results and discussion according to the suggestion of the reviewer and added headlines.

  1. The results regarding the prognosis should be explained better: was any multivariable analysis performed? Why vascular invasion is a confounder? Is E cadherin associated or not with prognosis?

Response: We believe the reviewer is referring to the VETC pattern and our analyses regarding vascular invasions. The question of additional statistical analyses raises an important point. We have performed these analyses and have now been able to show that in a multivariate analysis, the vascular invasion has a significant effect on survival in contrast to E-cadherin expression by immunohistochemistry. The association of E-cadherin with survival has also been carried out in the TCGA cohort (see figure 8, cf. comments for reviewers #3).

  1. The discussion regarding prognosis and prognostic factor should be reformulated in a more clearly, less repetitive and more incisive way (taking benefit of aims of the study).

Response: We thank the reviewer for this relevant assessment. The topic is complex, so a clear order is very important. We have now worked out the individual elements of our discussion and grouped the sections according to the topics. In addition, connecting paragraphs and new analyses have been added to improve readability.

Reviewer 2 Report

Gerber et al. reported that E:N-cadherin heterodi-38 mers are constitutively expressed in the hepatocytic lineage and only slightly altered in malignant  progression, especially in hemangionvasion, but do not fulfill the criteria for EMT in HCCs.

 Minor comment: "Both cadherins are well conserved among the species. The intracellular cadherin domain is linked to the actin cytoskeleton via the cytoplasmic plaque proteins α- and β-catenin (CTNNA1 and CTNNB1), plakoglobin (JUP), p120 (CTNND1), and vinculin (VCL)." How was the interaction between  "Adenomatous polyposis coli (APC)" and both cadherins? Does these cadherins have any effects on the anticancer treatments of HCC? Authors should discuss more in discussion section.

Author Response

Gerber et al. reported that E:N-cadherin heterodimers are constitutively expressed in the hepatocytic lineage and only slightly altered in malignant progression, especially in hemangionvasion, but do not fulfill the criteria for EMT in HCCs.

Minor comment: "Both cadherins are well conserved among the species. The intracellular cadherin domain is linked to the actin cytoskeleton via the cytoplasmic plaque proteins α- and β-catenin (CTNNA1 and CTNNB1), plakoglobin (JUP), p120 (CTNND1), and vinculin (VCL)." How was the interaction between "Adenomatous polyposis coli (APC)" and both cadherins? Does these cadherins have any effects on the anticancer treatments of HCC? Authors should discuss more in discussion section.

Response: We thank the reviewer for his important amendment. APC mutations have only been found in 3.2% of HCCs (www.cbioportal.org). A deeper analysis of the underlying pathways reveals that ß-catenin is of more general importance in HCCs (CTNNB1, TCGA cohort: mutations in 29.1% of cases), in accordance with the literature. We discuss the relevance of APC in more detail in the discussion and have included an additional figure concerning cadherins and catenins (novel figure 3). We had investigated APC in the liver in another study (Ueberham et al., Mol Cancer Res, 2015), yet we did not verify the association between APC and catenins in this study. We have added a paragraph on Wnt signaling, ß-catenin, and APC in the discussion.

Reviewer 3 Report

The manuscript  by Gerber et al aim to analyze the co-expression of E- and N-cadherin during the progression of hepatocelllar carcinoma in in vitro culture and in human samples. The impact on patients’ outcome and with other known markers have been also analyzed in in-house cohort as well as in the TCGA data set. This study follows a their previous discovery (2011) that the two cell-cell adhesion proteins were found concomitantly expressed in hepatocyte and hapatocellular carcinoma cell, the same models used in the present study. The data are interesting since in this particular cell types, normal and transformed, the co-expression of E- and N-cadherin seems unique. Here, the analyses performed in the in vitro model appear to add some more information about the adherens junctions AJs in these cells, however these data, that would really add some novel knowledges in the field of cell-cell adhesion, are poorly described and discussed. The fact that the complex E-/N-cadherin, is only present in TRITON-X100 containing lysates but it does not include α-catenin might mean that the relevant complex constitutes particular membrane structures different from stable AJs. Furthermore, the confocal immunofluorescenze (IF) could be shown different cell-cell sites of the relevant complex in in vitro culture and in human samples also showing the Z-projection. So, all these further analysis/description/comment should be added.

The epithelial-mesenchymal transition (EMT) occurring during some particular phases of carcinoma progression appears does not take place in this cells. However, at first stages, EMT can be considered a transcriptional reprogramming which involves the up-modulation of a class of transcription factors (TFs) which includes ZEB1; however, ZEB1 is only one of those TFs. Since these analyses have been performed in TCGA, at least the inverse correlation between E-cadherin and each of those TFs could be analyzed. If these analyses woukld not be successful then the Authors have some more evidence that EMT is not the mechanism activated during HCC progression. So far, the data do not really support the Authors’ conclusions.

The analyses on human samples seems interesting, but maybe the Authors could be better explain the scores calculated as presented in Table 1 since the data are not so comprehensible. What is H-score vs Score? And what is the column called ‘High’?

Minor:

Lanes 58 and 61: adhering junctions? I would use adherens junctions which can be abbreviated as AJs.

Materials and Method Section: The methods shoud appears under the appearance in the Results section.

P120 shoud be cited as p120 catenin, abbreviated as p120-ctn.

Lane 494: may be at least a reference is missed

Author Response

The manuscript by Gerber et al aim to analyze the co-expression of E- and N-cadherin during the progression of hepatocelllar carcinoma in in vitro culture and in human samples. The impact on patients’ outcome and with other known markers have been also analyzed in in-house cohort as well as in the TCGA data set. This study follows a their previous discovery (2011) that the two cell-cell adhesion proteins were found concomitantly expressed in hepatocyte and hapatocellular carcinoma cell, the same models used in the present study. The data are interesting since in this particular cell types, normal and transformed, the co-expression of E- and N-cadherin seems unique. Here, the analyses performed in the in vitro model appear to add some more information about the adherens junctions AJs in these cells, however these data, that would really add some novel knowledges in the field of cell-cell adhesion, are poorly described and discussed.

Response: Thank you very much for your interest in this manuscript and your critical amendments. In the revision, we have worked on the description in the discussion (see also the comments of Reviewer #1). In the frame of this study, our focus was on the relevance of E- and N-cadherin for homeostasis of normal liver and human hepato­carcino­genesis in situ, as in our previous work (Straub et al., 2011, J. Cell Biol.) focused on AJ of hepatocytes in in vitro models. Yet, we have taken up the comment of the reviewer and extended the discussion with this respect.

The fact that the complex E-/N-cadherin, is only present in TRITON-X100 containing lysates but it does not include α-catenin might mean that the relevant complex constitutes particular membrane structures different from stable AJs.

Response: The reviewer has made an important point. From our analyses, we conclude that cadherin-catenin-complexes are differentially solubilized with detergents. Obviously, the cadherin-complexes in the lipophilic cell membrane need more stringent detergents, whereas cadherin-catenin-complexes in the cytoplasm are more prone to be solubilized by milder detergents such as TRITON-X-100, although all complexes are detected with both buffer condition, yet in different amounts. From parallel immunofluorescence microscopy analyses comparing the localization of cadherins and catenins, we have no hint whatsoever that cadherin-catenin complexes may not be stable (compare novel figure 3).

Furthermore, the confocal immunofluorescenze (IF) could be shown different cell-cell sites of the relevant complex in in vitro culture and in human samples also showing the Z-projection. So, all these further analysis/description/comment should be added.

Response. In our previous study (Straub et al., J. Cell Biol, 2011), we concentrated on cultured hepatoma cells and analyzed E-N-cadherin complexes in more detail, also using immunoprecipitation and sucrose gradient analyses. In the present study, our focus is on hepatocarcinogenesis in situ. Z-projection is only possible in cultured cells, whereas in liver parenchyma, hepatocellular adenoma, and carcinoma in situ, sections always demonstrate different orientations of hepatocytes / hepatocytic tumor cells. For a frontal view of hepatocytes with view on apical / bile canalicular cell membrane, as well as lateral and basal cell membrane with orientation to the space of Disse, we now refer to the novel figure 3 (see insert in A’’ and explanatory comments in the figure legend). Z-projections in cell culture have not been analyzed so far, as the relationship of AJ to focal adhesions or other junction types was out of scope of the present study, and double-label immuno­fluores­cence microscopy was undertaken in methanol-acetone fixed cells to better preserve antige­nicity needed for complex interaction studies. Yet methanol-acetone-fixation naturally impairs the height of cells and therefore the Z-projection. If the reviewer feels, that Z-projection of culture cells is mandatory, we will try to work on E- and N-cadherin in relation to other junction complexes in paraformaldehyde-fixed cultured cells.

The epithelial-mesenchymal transition (EMT) occurring during some particular phases of carcinoma progression appears does not take place in this cells. However, at first stages, EMT can be considered a transcriptional reprogramming which involves the up-modulation of a class of transcription factors (TFs) which includes ZEB1; however, ZEB1 is only one of those TFs. Since these analyses have been performed in TCGA, at least the inverse correlation between E-cadherin and each of those TFs could be analyzed. If these analyses woukld not be successful then the Authors have some more evidence that EMT is not the mechanism activated during HCC progression. So far, the data do not really support the Authors’ conclusions.

Response. We believe that the reviewer raises an important point. Unfortunately, the concept of EMT is not well-circumscribed at all, in our literature search we identified over 40 relevant transcription factors and proteins that are thought to be directly related to EMT but none alone has been proven to be a gold standard for the presence of EMT. We had to set a focus here and concentrated on the mRNA of structural proteins and transcription factors with relevance to HCCs. Besides correlation analyses, which did not provide any additional insight, we performed an additional Cox regression analysis. This showed very clearly that CDH1 - in contrast to the univariate analysis - no longer has a significant influence on the survival of HCC patients. In addition, we were able to show that the overexpression of a few transcription factors (TCF3, CD151, and ZEB1) are decisive markers of an unfavorable prognosis in HCCs. Principally, the transcriptional pathways may not be completely separated from each other; as an example, the gene products of TWIST1 and TCF3 have been shown to be repressors of CDH1 expression. We have raised this issue as a possible cause of a link between CDH1 expression and survival in the discussion. Importantly, E- and N-cadherin are co-expressed in HCCs but are already constitutively expressed in normal hepatocytes. This contrasts with previous studies that associated changes in these two proteins with EMT but did not analyze normal tissue as a control in a statistically meaningful cohort. And we need to state, that this is in gross conflict with the notion of E- and N-cadherin being reregulated in carcinogenesis, as both are retained from normal to benign and malignant cells. Therefore, at least the presence of N-cadherin in HCCs does not fit the notion of EMT. The constitutive expression of N-cadherin in hepatocytes, as well as in cholangiocytes and respective tumors enables the use of N-cadherin as an immunohistochemical marker in the distinction of primary and secondary liver tumors (see also https://doi.org/10.3390/cancers14133091). We have now also emphasized this point in more detail in the revised version of the discussion.

The analyses on human samples seems interesting, but maybe the Authors could be better explain the scores calculated as presented in Table 1 since the data are not so comprehensible. What is H-score vs Score? And what is the column called ‘High’?

Response: This is a very important point, which we are happy to explain in more detail. For the best possible and correct assessment of E- and N-cadherin in hepatocarcinogenesis, we evaluated staining intensity as well manually as well digitally. H-score is a calculated value that is created using computer-assisted evaluation by the program QuPath. We describe that the score is determined by manual evaluation, which is more reproducible. Yet, the H-score allows a better quantification, but is more susceptible to variations in immune­histochemical staining strength and different dilutions, so it may vary between different laboratories. However, both values (H-score and manual score) correlate very well. So far, we have only commented on this in the methodology section. The dichotomization of expression patterns is important to understand our immunohistochemical results, as it is well reproducible and the software-based analyses are laborious and require digitized immuno­­histochemical section preparations. To facilitate the reader’s understanding here, we have addressed our methodological approach with an additional paragraph in the discussion and methods section.

Minor:

Lanes 58 and 61: adhering junctions? I would use adherens junctions which can be abbreviated as AJs.

Response: According to their function and molecular composition, cell-cell junctions are subclassified into the junction types tight junctions (sealing membranes), gap junctions (for communication), and adhering junctions (mechanical tethers). Besides adherens junctions (AJs), also other mechanically acting junction types fall in the group of adhering junctions, namely desmosomes, but also the so-called area composita junctions in cardiomyocytes. We have abbreviated adherens junctions accordingly (AJs).

Materials and Method Section: The methods shoud appears under the appearance in the Results section.

Response: The Materials and Methods Section appears in the usual order of the Journal Cells.

P120 shoud be cited as p120 catenin, abbreviated as p120-ctn.

Lane 494: may be at least a reference is missed

Response: We thank the reviewer for these valuable comments and have corrected the two issues.

Round 2

Reviewer 1 Report

-

Author Response

Thank you very much for your final assessment of our revised manuscript. We have improved English language, and worked on a better presentation of results and discussion.

Author Response

Point to point response to the reviewer 3:

I still think that the description of the IP-derived data is not well described and the conclusions are still not correct. My impression is the there are two different pool of N-cadherin and only one is in complex with E-cadherin. It seems that the AJs by E-cadh/beta catenin can contain a little bit of N-cadh, the latest being in another membrane compartment. Furthemore, three different buffers are used for the IP. When didi you use the ‘Empigen-IP-Buffer’? Which buffer did use for the human sample of Fig.A2? Which was the the unrelated AB used? Can you a petter wetsrn blo including the control in the same experiment?

Response: We thank the reviewer for his thorough assessment which we are happy to take up and answer in-depth, please compare also the revised version of the manuscript. We used 3 (and more) different buffers for immunoprecipitation. Empigen-containing IP buffer yielded less protein complexes, so after some first tests, we only used Triton-X-100 IP buffer and RIPA buffer as demonstrated in figure 2 (in PLC cells) as well as appendix figure A2 (in human HCC tissue: RIPA-buffer). For all immunoprecipitation experiments, immunoprecipitation has been performed also in parallel with an unrelated antibody of the same species, namely a mouse myeloma antibody as well as with precleared beads without antibodies each of which did not precipitate cadherin or catenin-complexes at all. Furthermore, in another project, we used antibodies against perilipins but no cadherins or catenins were coprecipitated either. For further information, see also elaborate immuno­precipitation experiments in the previous publication (Straub et al., 2011). As disparate E- and N-cadherin fractions were detected in PLC cells only, but not in human liver or HCC tissue, we investigated cadherins also with double label immunofluorescence microscopy. In PLC-cells, besides colocalized / obviously heterodimeric E:N-cadherin complexes also homodimeric E- and/ or N-cadherin complexes were detected at some membrane areas, yet, again, in liver and HCC tissue, only complete colocalization of E- and N-cadherin was detected indicative of heterodimeric E:N-cadherin complexes. In this regard, we postulate, that cultured cells harbor an intrinsic heterogeneity of E- and N-cadherin-containing AJ, whereas the AJ in tissue contain coalescent complexes.

By the multivariate analysis ZEB1 emerged very clearly as an independent prognostic factor; indeed, ZEB1 is one of the major transcription factor guiding the cadherin switch and EMT. Maybe you intended the ‘cadherin swtch’ which is related to the acquisition of a more motile phenotype. Furthermore, the increased of N-cadherin at the intra-hepatic sites of VETC is also observed and these sites are the ones related to metastases. Isn’t it? Can be possible that the cadherin switch, typical of EMT, occur also in HCC but only at the sites where the diffusion of cancer cells occur?

Can you a little revise the Conclusions in lanes 841-44? Accordingly, in the Abstract.

Response: Thank you very much for your comment on the significance of ZEB1 in regulating EMT and the cadherin switch. We have two comments on that issue. First, although the expression of N-cadherin is significant, it is only minimally increased. The minimal difference is only apparent in the very large cohort we analyzed. Second, as described in the manuscript, there are alternative plausible explanatory models for these minimal N-cadherin differences. We think that the relatively increased vascular density in VETC HCC is the key factor for the increased N-cadherin expression and not EMT. The concept of EMT presents the cadherin switch as a local effect present only in the invasive front and in tumor lymhangio- and hemangioinvasion. Yet, we did not observe changes in E- or N-cadherin expression at these sites. However, this point is important, and we have discussed it further in the manuscript.

There was a misunderstanding: the sequence of the methods should follow the appearance in the Results section.

The manuscript still needs an English revision. See for example lanes 390-92 but other flaws are all along the manuscript.

Response: Thank you for your comments. We have changed the manuscript accordingly and it has been proofread by an experienced English speaker.
